# Seasonal Trends in Cardiac Troponin I Concentration and Creatine Kinase and Aspartate Aminotransferase Enzyme Activity in Relation to Myocardial Velocity Rates in Eventing Horses

**DOI:** 10.3390/ani15213198

**Published:** 2025-11-03

**Authors:** Insa Rump-Dierig, Johanna Giers, Charlotte Frenzel, Sabita Stöckle, Heidrun Gehlen

**Affiliations:** 1German Olympic Committee for Equestrian Sports, 48231 Warendorf, Germany; insa_rump@web.de; 2Equine Clinic: Surgery and Radiology, Freie Universität Berlin, 14163 Berlin, Germany

**Keywords:** fatigue, eventing, performance diagnostic

## Abstract

**Simple Summary:**

In this study, eventing horses were monitored over the course of a competition season. This study examined the extent to which their skeletal and cardiac muscles were stressed by training and competition. After riding, creatine kinase (CK) and aspartate aminotransferase (AST), as well as cardiac troponin I (cTnI), levels rose briefly but mostly returned to normal. Echocardiographic measurements demonstrated that myocardial velocities remained largely stable but showed slight seasonal fluctuations. A significant correlation was found between cTnI levels and early diastolic myocardial velocity (Em). We conclude that blood tests and cardiac ultrasound can be used to determine whether a horse is overexerting itself. Therefore, regular check-ups help to better protect the health and performance of horses.

**Abstract:**

This study examines seasonal changes in muscle and heart parameters in eventing horses over the course of a competition season. Blood levels of the enzymes creatine kinase (CK) and aspartate aminotransferase (AST) as well as the heart muscle-specific concentration of cardiac troponin I (cTnI) were measured before (pre), 30 min (p30) and 24 h (p24) after competitions. Creatine kinase (CK: median pre-competition = 175 U/L, 30 min post = 221 U/L, 24 h post = 140 U/L), aspartate aminotransferase (AST: pre = 319 U/L, p30 = 335 U/L, p24 = 333 U/L), and cardiac troponin I concentration (cTnI: pre = 0.006 ng/mL, p30 = 0.011 ng/mL, p24 = 0.007 ng/mL) exhibited partial normalization by 24 h post-exercise but at the same time demonstrated significant seasonal variation (*p* < 0.001). Echocardiographic assessments revealed sustained high-level myocardial velocities, with occasional modest seasonal declines. A significant correlation was identified between cTnI levels and early diastolic myocardial velocity (Em) (Spearman’s Rho: pre-exercise 0.323, Rho p30: 0.357), whereas a negative correlation was manifest at 24 h (Spearman’s Rho = −0.300). These findings suggest a heightened sensitivity of diastolic myocardial velocity to myocardial injury. Given that diastolic dysfunction frequently constitutes an early manifestation of myocardial compromise, our results underscore the utility of biomarkers alongside myocardial velocity measures as valuable tools for the early detection of subclinical fatigue in high-performance sport horses.

## 1. Introduction

Creatine kinase (CK) and aspartate aminotransferase (AST) are enzymes that are expressed in both skeletal muscle and cardiac muscle. In equestrian sports, they are used, among other parameters, to measure the performance intensity of the muscles [1,2,3]. Enzyme activities are influenced by various factors: younger horses tend to show a higher increase in CK enzyme activity after exercise than older animals [4], and enzyme activities can also be influenced by the season [5]. In horses, CK has a half-life of 123 min and reaches its peak approximately 12 h after exercise [6]. Enzyme activity returns to normal levels within 48 h. AST enzyme activity peaks approximately 24–48 h after exercise in horses and remains elevated for up to 8 days [7].

Cardiac troponin I (cTnI) is a protein that is expressed exclusively in heart muscle cells and is released when the heart muscle is damaged. As a component of the contractile unit of the muscle, phosphorylated cardiac troponin causes calcium to be released more quickly, allowing the heart muscle to relax more rapidly. Therefore, an elevated cardiac troponin concentration in the blood indicates damage to the heart muscle, whereas damage to the skeletal muscle does not lead to an increase in cardiac troponin I concentration [8]. In mammals, the concentration of cTnI reaches its peak within a few hours after exercise [2,8,9], although it should be noted that structural changes in the heart can also lead to a permanent increase in concentration [10]. Internationally, a value of ≥0.03 ng/mL is being discussed as the cut-off value in horses above which damage to the heart muscle can be assumed [10,11,12].

Studies in human medicine have shown that heart rate has a significant influence on myocardial velocities in both ventricles. This is also evident in various physiological mechanisms and adaptations. In mammals, as the heart rate increases, diastole shortens, which means that there is less time available for the ventricles to fill [13]. This can affect myocardial velocity during systole, as the ventricles may not be completely filled, which in turn can lead to reduced contractile force. In contrast, a higher heart rate can lead to increased preload of the myocardium through the Frank-Starling mechanism, which increases the contractile force. As the afterload decreases with the increased heart rate, the Frank-Starling mechanism increases the volume with increasing heart rate. The higher volume leads to greater stretching of the heart muscle fibers. The contractile force increases in turn because greater stretching of the heart muscle fibers leads to better overlap of actin and myosin filaments, which increases myocardial velocity during systole [12]. Despite a shorter diastole, sufficient volume is thus maintained.

While both ventricles are affected by these effects, they show different responses to changes in heart rate. The left ventricle, which is responsible for systolic blood pressure, has to adapt more and can achieve higher myocardial velocity at high frequencies than the right ventricle, which operates at lower pressure. However, the right ventricle exhibits higher overall myocardial velocities than the left ventricle [14]. Myocardial velocity in horses over the course of the season is therefore an important indicator of cardiac performance and adaptation to training.

Over the course of a tournament season and with ongoing training, the heart muscle of a horse undergoes adjustments: the maximum heart rate remains largely unchanged, but the heart rate during submaximal exercise decreases [15,16]. The stroke volume increases, which leads to improved pumping performance of the heart [14]. The ejection time (ET) lengthens, while the pre-ejection period (PEP) decreases, indicating more efficient heart contraction [17,18]. Towards the end of the season, some horses may show signs of cardiac fatigue, which can manifest itself in reduced systolic and/or diastolic function [16,17]. Factors influencing myocardial velocity are the age and sex of the horse: mares tend to have higher heart rates than stallions or geldings [16].

Monitoring myocardial velocity throughout the season can provide valuable insights into the horse’s training adaptation and performance: regular echocardiographic examinations enable the early detection of overtraining or cardiac fatigue. Improved myocardial velocity and cardiac efficiency often correlate with successful competition participation [18].

Elevated cardiac troponin levels are often associated with reduced systolic myocardial velocity in human medicine [19], indicating impaired contractility of the heart muscle. In patients with heart failure, studies show that an increase in cardiac troponin concentrations often correlates with a decrease in diastolic myocardial velocity, indicating impaired relaxation of the heart [20,21].

Troponin levels in horses can also be outside the reference interval due to their training, as several studies have already shown [2,22,23,24]. However, the values should return to their baseline within 24 h on average [2,25]. In addition, prolonged physical exertion in horses has been shown to be associated with impaired diastolic function of the left ventricle, which filled only partially for 7–12 h [26]. Nevertheless, the biochemical indicators for hydration status normalized within these 7–12 h, suggesting that the observed changes are not exclusively attributable to altered preload conditions [25].

To date, no correlation has been found between myocardial velocity and cardiac troponin in any mammal [26]. This is explained by the fact that cTnI elevations reflect acute stress or damage to the myocardium, while reduced myocardial velocities indicate impaired cardiac function. In addition, cardiac troponin reaches its peak value earlier (2–6 h after stress) than the sustained reduction in myocardial velocity (up to 21 h).

Nevertheless, the aim of this study was to track the development of the parameters over the course of an eventing season and to gain a more comprehensive insight into cardiac recovery after intense stress. It was hypothesized that CK, AST and cTnI would increase over the course of the season and that changes in myocardial velocities would correlate with cardiac troponin levels.

There is currently no evaluation of the course of an eventing season for the parameters mentioned here in high-performance riding horses. The development of the values under prolonged, repetitive exertion could provide important insights into the recovery capacity of our sport horses and thus contribute to their optimal training intensity.

## 2. Materials and Methods

The data collected originated from the 2022 recreational monitoring field study by Johanna Giers, Katharina Böhm, Charlotte Frenzel and Insa Rump-Dierig. In this study, the enzyme activities of CK, AST and cTnI concentration, as well as ultrasound measurements of systolic and diastolic myocardial velocity of the left ventricle and interventricular septum were measured. Testing times were in the morning before the cross-country course (pre-sample), 30 min after crossing the finish line and 24 h after the pre-sample in 19 eventing horses. The progression of the parameters over the 2022 competition season was examined. Furthermore, seasonal fluctuations in the parameters and a possible correlation with the progression over the competition season were investigated.

Riders and owners gave their written informed consent to participate voluntarily and free of charge. The study was registered with the Berlin State Office for Consumer Protection (1-02.04.40.2022.VG006), but was not classified as animal testing. The blood samples were taken by veterinarians who were involved in routine performance diagnostics as part of the performance monitoring program of the German Olympic Equestrian Committee (DOKR). All horses were clinically and echocardiographically examined by veterinarians from the DOKR project and declared healthy before participation.

### 2.1. Study Design

A total of eight riders with 19 horses participated in the study. The investigations took place between January and September 2022 at 14 international eventing tournaments in the 2- to 4-star category at six different venues in Germany and Poland (Section A.1: Table A1).

All horses were sampled before the start of the season (baseline testing in January) and at two to five tournaments during the season, resulting in one pre-season sample plus four to 16 tournament samples from each horse. At each competition, the horses underwent the prescribed veterinary checks before and after the cross-country course, during which the horses were declared ‘fit to compete’.

The horses participating in the study were between seven and 15 years old, with an average age of 11. The population consisted of ten mares and nine geldings. The horses belonged to nine different warmblood breeds. The relevant information for identifying the horses was taken from the FEI database.

The riders were between 21 and 39 years old, with an average age of 28. Three male and five female riders participated in the study.

In accordance with the rules of the Fédération Equestre Internationale (FEI) for international eventing competitions, 9 rides at the two-star level, 31 rides at the three-star level and 15 rides at the four-star level were included in the sample. The average score in the cross-country test was 9.2 penalty points. In two rides, the riders were eliminated due to falls, and in one ride, the rider withdrew.

### 2.2. Blood Samples

Baseline samples were taken in January before any horse went to a horse show. Tournament samples were taken in the morning before the cross-country course (pre), thirty minutes (p30) after the end of the ride, and 24 h (p24) after the pre sample the next morning. The samples before the ride and the next morning were taken between 4:00 a.m. and 7:30 a.m. Due to different start times, the p30 samples were taken between 9:00 a.m. and 5:00 p.m. The time span between p30 and the p24 sample ranged from 11 to 21 h, while the time span between the pre-sample and the p24 sample was 24 h (+/−1 h).

Blood samples were taken from the jugular veins of the horses, with the puncture site disinfected with 1-propanol. Venous blood was collected using a Vacutainer system with 20-G needles and PET (polyethylene terephthalate) tubes.

After each blood collection, the EDTA whole blood tubes were immediately cooled to +5 °C, while the serum gel tubes were stored at room temperature for 30 to 60 min until coagulation was complete. A portable centrifuge, type EBA 200 from Andreas Hettich GmbH & Co. KG (Tuttlingen, Germany), was used to centrifuge the serum tubes for 10 min at 1000× *g* before transferring the serum to uncoated plastic tubes and immediately freezing it at −20 °C.

Hematology was examined using laser flow cytometry and laminar flow impedance in a ProCyte DX hematology analyser from IDEXX Laboratories Inc. (Westbrook, ME, USA). The blinded serum samples were further processed in the testing laboratory (LABOKLIN, Bad Kissingen, Germany). cTnI was measured using a chemiluminescence assay (LIA) with an ADVIA Centaur XPT 2000 (Siemens, Munich, Germany). All other blood chemistry values were measured photometrically (PHO) or potentiometrically (POT) with a Cobas 8000 analyser (Roche, Basel, Switzerland).

A complete set of samples is available for 51/55 rides plus the baseline samples. Incomplete sample sets were removed from the statistics.

### 2.3. Echocardiography

The echocardiographic examinations were performed simultaneously to the blood samples as a baseline examination in January and as tournament examinations before the cross-country course (pre), thirty minutes (p30) after the end of the ride and 24 h (p24) after the pre sample the next morning. In all examinations the portable ultrasound device ‘Vivid i’ from GE Medical Systems with a 3 s transducer and simultaneous ECG recording was used. The frequency was 1.7/3.4 megahertz. A frame rate of at least 50 frames/second was ensured. All echocardiographic examinations were performed by Charlotte Frenzel herself, since different handling would alter the results. During the ultrasound examination, an electrocardiogram was recorded simultaneously to determine the heart rate. The recordings were then evaluated offline using EchoPAC software (version 110.1.1) from GE Healthcare.

During the baseline examination in January before the tournament season, a general and special echocardiographic examination was performed in B-mode and M-mode. The general echocardiographic examination was used to assess the heart. Colour Doppler was used to rule out significant valve insufficiencies.

Myocardial velocity was measured using tissue Doppler (PW-TDI) at rest. The left ventricle was imaged in cross-section from the right side at the level of the chordae tendinae, in the so-called short axis. The penetration depth was 28 cm, and the image width was reduced to such an extent that the myocardium could still be completely captured. The main focus was on the left ventricular free wall, and the measurement gate (sample volume) was positioned centrally.

In the eventing competitions with different levels of difficulty (Concours Complet Internationale (CCI) ** to ****), only echocardiographic tissue Doppler examinations were performed. Since not every horse was used in every event, each horse was examined two to six additional times. An examination cycle consisted of a resting examination in the morning before the cross-country event, 30 min and 24 h after the event.

During offline evaluation of the tissue Doppler measurements (PW-TDI), the maximum systolic myocardial velocity (Sm), the maximum early diastolic myocardial velocity (Em) and the maximum late diastolic myocardial velocity (Am) were determined using the ECG. In the left ventricular free wall, the systolic velocities had a positive sign and the diastolic velocities had a negative sign. Three measurements were taken in each case and the mean value was calculated from these.

Of 201 tissue Doppler measurements, 187 were of sufficient quality to perform and evaluate the measurements.

### 2.4. Data Analysis

Statistical analyses were performed using Jamovi (version 2.3.21.0) and IBM SPSS Statistics (version 30.0.0). The normal distribution of the data was assessed visually using box plots, histograms and Q-Q plots. The Pearson correlation coefficient was used to investigate the relationship between myocardial velocities and measured troponin levels (cTnI). To investigate the influence of the season on myocardial velocities and troponin levels, linear mixed models were used with the season as a fixed effect and the horse at the tournament as random effects. The season was divided into four time periods according to the seasons: 0 = baseline samples in winter without subsequent exercise, 1 = spring (March and April), 2 = summer (May to mid-August), 3 = autumn (mid-August to October).

## 3. Results

### 3.1. Values over the Course of the Season

The activity of the muscle-specific enzymes CK and AST showed significant seasonal variation. CK activity increased significantly (*p* ≤ 0.001) at all measurement points over the course of the season. In addition, a gradual decrease in peak values was observed. AST enzyme activity values increased significantly (*p* ≤ 0.001) at all measurement points over the course of the season. Cardiac Troponin I increased slightly over the course of the season and returned to its pre-season baseline value at the end of the season (*p* ≤ 0.001–0.97). The average AST value was 241 U/I, while CK had an average value of 167 U/I and cTnI had an average value of 0.037 ng/mL (Table 1).

#### 3.1.1. Creatine Kinase (CK)

CK (U/I) showed a significant increase (*p* ≤ 0.001) over the course of the season at all measurement points, most strongly at measurement point p30 (Figure 1). The pre-value at the end of the season was on average comparable to the p24h value for season 2. The peak values were particularly high in season 2 and decreased again at the end of the season.

#### 3.1.2. Aspartate Aminotransferase

AST (U/I) showed a significant increase (*p* ≤ 0.001) over the course of the season at all measurement points (Figure 2). The measurement points within a season did not differ significantly from one another.

#### 3.1.3. Cardiac Troponin I

Cardiac Troponin I (cTnI; ng/mL) rose slightly over the course of the season and fell back to the baseline season’s value at the end of the whole season (*p* ≤ 0.001–0.97; Figure 3). The peak values were highest at all measurement times in season 1, while no high peak values were recorded in season 3.

#### 3.1.4. Myocardial Velocities

Three different focuses were set for observing the myocardial velocity of the interventricular septum (IVS) and the left free wall (left wall/lw): Am = maximum late diastolic myocardial velocity, Em = maximum early diastolic myocardial velocity, Sm = maximum systolic myocardial velocity (Table 2 and Table 3).

When viewed seasonally, there were only minor differences in myocardial velocities, with only individual seasons showing significant differences from each other: lw Sm, season 1–2, *p* < 0.001; lw Em, season 1–3, *p* < 0.001; IVS Em, season 1–2, *p* < 0.001).

The left ventricle showed a slight adaptation by increasing systolic velocities (Sm), indicating improved pumping function. Towards the end of the season, slight decreases in diastolic velocities (Am and Em) were observed in some horses (Figure 4).

### 3.2. Correlation of Myocardial Velocities with Cardiac Troponin I

The analysis of the correlation between myocardial velocities and cTnI values using heat maps and Spearman’s Rho (scc, Section A.2: Table A2) showed a moderate significant correlation between cTnI values and early diastolic myocardial velocity Em of the left wall (lw) at all measurement points. The measurement taken before exercise (time point pre, Figure 5a) showed a positive, moderate, significant correlation of Spearman’s Rho (scc) = 0.323; *p* = 0.024. Half an hour after crossing the finish line (p30, Figure 5b), a positive, moderate, significant correlation of Spearman’s Rho (scc) = 0.357; *p* = 0.013 was obtained.

In addition, a negative, moderate, significant correlation was found between the cTnI values and the early diastolic myocardial velocity Em of the left wall after 24 h (p24, Figure 5c) of scc = −0.3; *p* = 0.04. The other myocardial velocities of the pre-samples, the p30min values and the p24h values did not correlate significantly with cTnI.

## 4. Discussion

### 4.1. Seasonal Progression

In horses, CK and AST typically increase 4–6 h (CK) and 24–48 h (AST) after intense exercise [2,7,27]. Cardiac Troponin I reaches peak values approximately 2–6 h after exercise and returns to normal within 12–24 h [6,9,28]. There is currently no evidence of a cumulative increase in these values over the course of a sporting season in horses. Human research has already shown that CK and AST levels can accumulate during a sporting season [29]. However, human medical research teams are still investigating the causes of an increase in cTnI levels during prolonged physical exertion. The cause remains unknown [30].

Giers et al. [2] showed that CK, AST and cTnI can exceed reference values after cross-country training sessions for eventing horses. The results shown here revealed a significant increase in the average CK and AST values measured over the whole season. In contrast, the average cTnI value rose only briefly and to a much lesser extent and also fell again before the end of the season. At the same time, it can be observed that particularly high cTnI values occur less frequently during the season. Peaks in stress may therefore have less impact on the heart muscle in these horses, suggesting a better overall constitution of the heart muscle.

Various factors could lead to a cumulative increase in blood values: on the one hand, rising temperatures over the course of the season could place greater strain on the horses and could be a limiting factor to this study. However, Munsters et al. [31] were able to show that horses can adapt to increased temperatures and humidity within 14 days in terms of their lactate levels. Unfortunately, lactate levels were not compared in this study. On the other hand, more intensive training and competition phases could lead to repeated microtrauma in the muscles and thus result in an increase in blood values, as CK and AST are released into the blood in the event of muscle injuries, cramps or muscle soreness. In addition, insufficient recovery between intense periods of exertion can cause enzymes to accumulate [32]. The fact that cTnI did not increase proportionally to AST and CK levels in this study is probably due to the different origins of the enzymes. AST and CK are both cardiac and skeletal muscle enzymes, while cTnI is a specific heart muscle marker. Accordingly, the release of enzymes from the skeletal and cardiac muscles may be increased, while the cardiac-specific cTnI concentration did not increase. Therefore, it can be concluded that the skeletal muscles of the horses were strained, but not the heart muscles. Furthermore, it should be noted that the blood samples were not measured at the same time as the expected peaks of the respective parameters, so a certain deviation from the actual course is quite likely.

The heart rate influences myocardial velocities by shortening the diastole, increasing contractile force via the Frank-Starling mechanism, and altering the duration of the action potential. These factors have different effects on the left and right ventricles and can therefore influence the efficiency and function of the heart under different physiological conditions.

Myocardial velocities, with a focus on the interventricular septum and the left free wall, showed no significant changes over the course of the season. Looking at the individual calendar weeks, a peak in myocardial velocities was observed in calendar week 15. Since the rides were measured at different equestrian events with different courses, it can be assumed that the altitude of the cross-country course had an influence, which can have a significant impact on the intensity of the course. In addition, slight variations in the handling of the ultrasound measurement can also lead to altered measurement values, as can stress and excitement during the measurement [33].

### 4.2. Correlation Between Troponin and Myocardial Velocities

The results of this study show that only early diastolic myocardial velocity (Em) correlates with the measured troponin values (cTnI). Neither systolic nor late diastolic myocardial velocity shows a significant correlation with troponin values. This could indicate that these phases of the cardiac cycle are less sensitive to changes caused by myocardial damage or that they are more strongly influenced by other hemodynamic factors. The correlation of Em, in turn, suggests that early diastolic myocardial function is more sensitive to structural damage, as indicated by elevated troponin levels [34]. An increase in troponin levels appears to particularly impair early myocardial relaxation, resulting in a reduced Em velocity. This observation is consistent with previous studies in human medicine that have shown that diastolic dysfunction is often an early manifestation of myocardial damage [35,36]. The positive correlation between troponin and myocardial velocity before exercise and at p30min, and the negative correlation at p24h, can possibly be explained by the fact that the sampling times do not coincide with the expected peak troponin concentrations. The maximum troponin concentration is expected to be between 2 and 6 h after exercise, while 24 h after exercise, the concentration is significantly reduced or may even have fallen to the baseline value [2,25].

Other parameters of cardiac function, such as left ventricular filling pressure or preload, could provide additional information about the relationship between myocardial velocities and troponin levels, but these were not considered in this study.

It should be noted that the study is based on a relatively small sample size, which could distort the results. In addition, not all horses were sampled and examined equally, which also limits the results of this study.

## 5. Conclusions

This study showed that skeletal muscle enzymes such as CK and AST can rise above reference values during a competition season, even without the horse being subjected to stress immediately prior to blood sampling. The duration of the rest period required for these parameters to return reliably to within the reference values remains to be determined. The observed increase in troponin during the season indicates the presence of cardiac fatigue in eventing horses, but no myocardial damage was detected. Significantly elevated cTnI levels, as would be expected in myocardial damage, were observed even less frequently during the course of the season.

In this study, a correlation was found between early diastolic myocardial velocities and measured troponin levels. However, the results showed no correlation with other heart rhythm parameters, which may indicate that these two parameters measure different aspects of heart function and damage and should possibly be evaluated independently of each other in clinical practice. Further studies are needed to better understand the exact relationship between these parameters and to clarify which additional factors influence myocardial velocities.

In summary, myocardial velocity in horses undergoes dynamic changes over the course of the season that are closely related to training status and competition stress. Careful monitoring and interpretation of these parameters can contribute to improved performance and health maintenance in sport horses. To assess whether troponin elevations indicate a pathological condition, further measurements after exercise and echocardiography are necessary.

## Figures and Tables

**Figure 1 animals-15-03198-f001:**
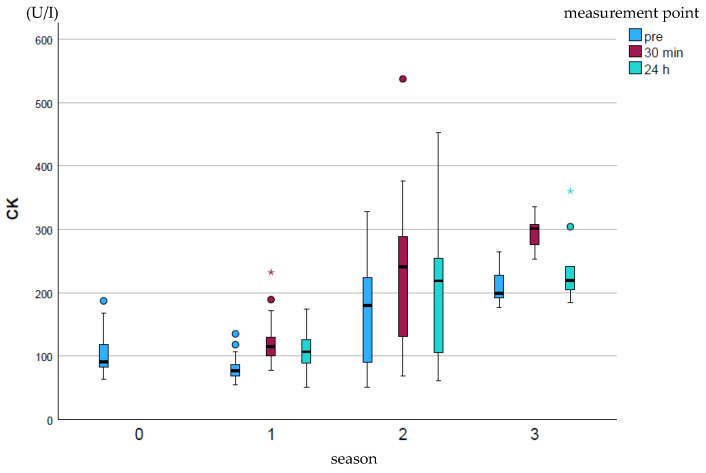
Development of CK enzyme activity over the course of the season; 0 = baseline testing (January), 1 = spring (March and April), 2 = summer (may to mid-August), 3 = autumn (mid-August to October) according to the measurement times pre (blue), p30min (red) and p24h (green), n = 175. Dot (circle): Represents a slight outlier. These data points are between 1.5 and 3 times the interquartile range away from the box. Star (*): Represents an stronger outlier. These data points are more than 3 times the interquartile range away from the box.

**Figure 2 animals-15-03198-f002:**
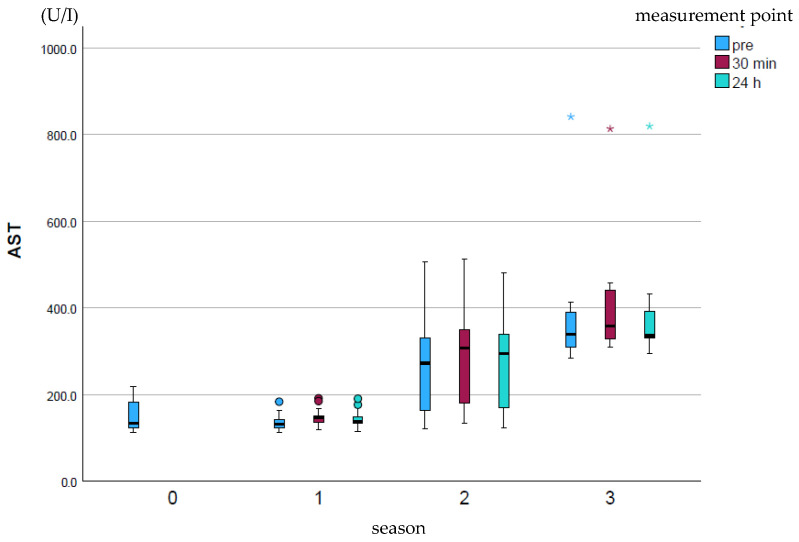
Development of AST enzyme activity over the course of the season; 0 = baseline testing (January), 1 = spring (March and April), 2 = summer (May to mid-August), 3 = autumn (mid-August to October) according to the measurement times pre (blue), p30min (red) and p24h (green), n = 175. Dot (circle): Represents a slight outlier. These data points are between 1.5 and 3 times the interquartile range away from the box. Star (*): Represents an stronger outlier. These data points are more than 3 times the interquartile range away from the box.

**Figure 3 animals-15-03198-f003:**
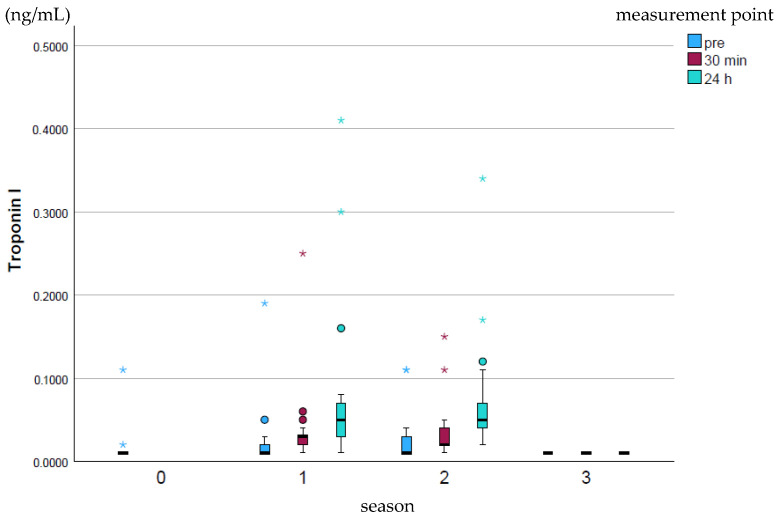
Development of cTnI values over the course of the season; 0 = baseline testing (January), 1 = spring (March and April), 2 = summer (May to mid-August), 3 = autumn (mid-August to October) according to the measurement times pre (blue), p30min (red) and p24h (green), n = 175. Dot (circle): Represents a slight outlier. These data points are between 1.5 and 3 times the interquartile range away from the box. Star (*): Represents an stronger outlier. These data points are more than 3 times the interquartile range away from the box.

**Figure 4 animals-15-03198-f004:**
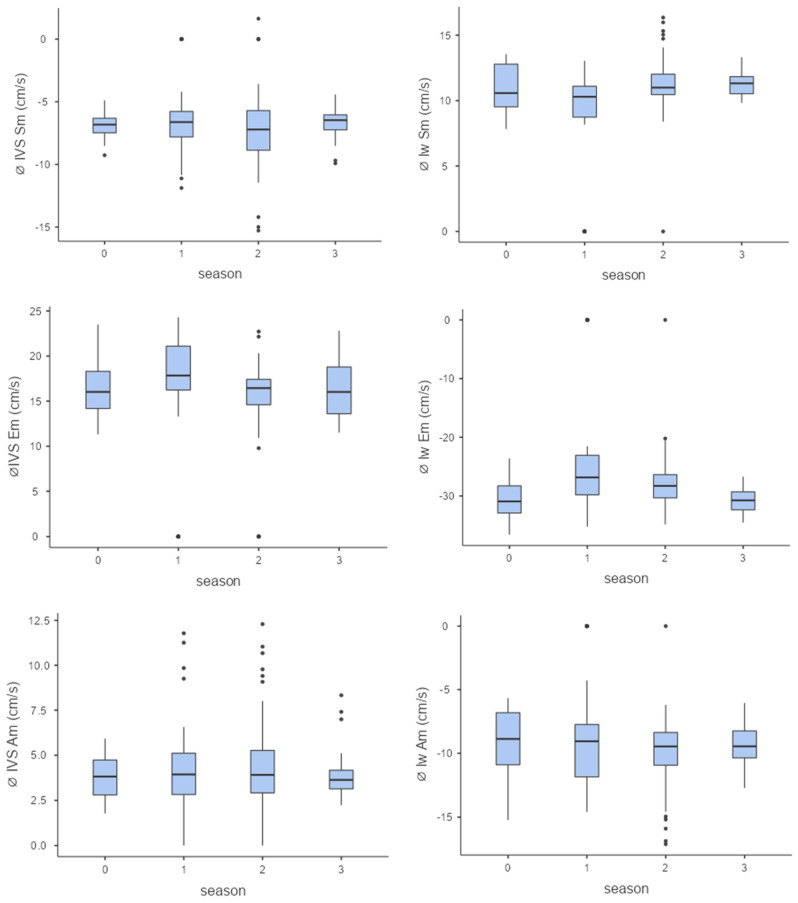
Myocardial velocities (from top to bottom Sm, Em, Am, Sm = maximum systolic myocardial velocity, Em = maximum early diastolic myocardial velocity, Am = maximum late diastolic myocardial velocity) of the interventricular septum (left column) and the left free wall (right column) over the course of the season (0 = baseline testing (January), 1 = spring (March and April), 2 = summer (May to mid-August), 3 = autumn (mid-August to October, n = 176).

**Figure 5 animals-15-03198-f005:**
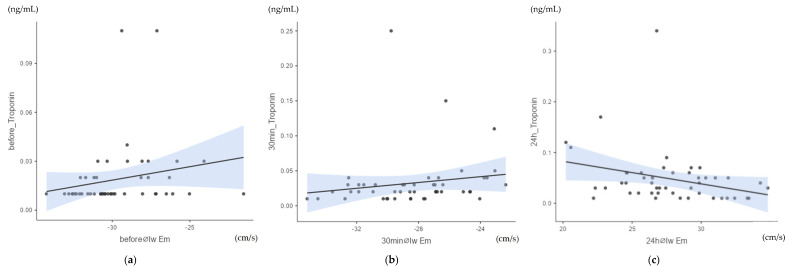
(**a**) Correlation between cTnI values and early diastolic myocardial velocity Em of the left wall (lw) before exercise (pre): Spearman’s Rho (scc) = 0.323; *p* = 0.024; n = 48. (**b**) Correlation between cTnI values and early diastolic myocardial velocity Em of the left wall (lw) after 30 min (p30): Spearman’s Rho (scc) = 0.370; *p* = 0.013; n = 48. (**c**) Correlation between cTnI values and early diastolic myocardial velocity Em of the left wall (lw) after 24 h (p24): Spearman’s Rho (scc) = −0.300; *p* = 0.04; n = 48.

**Table 1 animals-15-03198-t001:** Key figures for blood values of cTnI, AST and CK as the mean values across all measurement times.

	AST (U/I)	CK (U/I)	cTroponin I (ng/mL)
n	175	175	175
missing	0	0	0
mean value	241	167	0.037
median	183	153	0.020
standard deviation	133	88,5	0.055
minimum	112	51	0.010
maximum	841	537	0.410

**Table 2 animals-15-03198-t002:** Descriptive statistics: Key figures for myocardial velocities of the left free wall as the mean values across all measurement times.

	⌀ lw HR (cm/s)	⌀ lw Sm (cm/s)	⌀ lw Em (cm/s)	⌀ lw Am (cm/s)
n	176	176	176	176
missing	62	62	62	62
mean value	35.20	10.40	−27.10	−9.26
median	34.20	10.70	−28.50	−9.32
standard deviation	12.30	2.94	7.410	3.28
minimum	0.00	0.00	−36.60	−17.10
maximum	73.70	16.30	0.00	0.00

* lw = left wall, HR = heart rate, Sm = maximum systolic myocardial velocity, Em = maximum early diastolic myocardial velocity, Am = maximum late diastolic myocardial velocity.

**Table 3 animals-15-03198-t003:** Descriptive statistics: Key figures for myocardial velocities of the interventricular septum as the mean values across all measurement times.

	⌀ IVS HR (cm/s)	⌀ IVS Sm (cm/s)	⌀IVS Em (cm/s)	⌀ IVS Am (cm/s)
n	176	176	176	176
missing	62	62	62	62
mean value	34.60	−6.78	16.10	4.18
median	33.70	−6.83	16.80	3.78
standard deviation	12.50	2.61	5.08	2.28
minimum	0.00	−15.30	0.00	0.00
maximum	75.30	1.63	24.30	12.30

* IVS = interventricular septum, HR = heart rate, Sm = maximum systolic myocardial velocity, Em = maximum early diastolic myocardial velocity, Am = maximum late diastolic myocardial velocity.

## Data Availability

The data presented in this study are available on request from the corresponding author. The data are not publicly available due to privacy.

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
