# Peer review of "Seasonal Trends in Cardiac Troponin I Concentration and Creatine Kinase and Aspartate Aminotransferase Enzyme Activity in Relation to Myocardial Velocity Rates in Eventing Horses"

_animals, 2025, doi:10.3390/ani15213198_

Round 1
Reviewer 1 Report
Comments and Suggestions for Authors
To the authors: Thank you for your contribution of this well written article. The article investigates overall changes to AST, CK, Cardiac Troponin and velocity over the course of the show season. I do have a few minor suggestions prior to publication.
Methods:
Was each individual horse imaged by the same echocardiographer at each timepoint? As even hand angle can change your velocity, I would assume different clinicians imaging may contribute to differences.
For the cardiac troponin, has this analyzer been validated in the horse? I believe there can be issues with accurate measurements using the IMMULITE 2000 which is the same technology.
Results:
Was there any factor of time between shows in the results calculation? Later in the season these shows tend to become more back to back as people are trying to make year-end qualifiers/move up for the next season.
Discussion:
In the seasonal progression was the temperature considered as part of the reason AST and CK were elevated? It looks like these were elevated in the warmer months where you would overall expect more exertion due to increased ambient temperature. Did you look at the overall TPR of the horses?
Reviewer 2 Report
Comments and Suggestions for Authors
Reviewer comments for manuscript animals-3914895
General comments: Overall, an interesting paper. I think the manuscript needs more work before it is ready for publication, but it is a good start.
Specific comments:
The simple summary is not simple enough, and the abstract is missing key parts. Please look at recently published Animals articles for examples from other authors, as well as review the guidelines for authors set by this journal, and rewrite both. As they currently stand, both the simple summary and the abstract are unacceptable.
Line 50-53: are these values specific to the horse, or to mammals, or to humans? Please clarify.
Line 54-55: I would also include mention of the regulatory component of troponin as it relates to muscle contractions. Calling it merely a structural protein seems a bit simplified.
Line 57: if you mean skeletal troponin I (and not cardiac troponin I), you should state that and not just say “troponin I”. please correct
Line 57-61: are these values specific to the horses, or for all mammals? Please clarify.
Line 62-120: throughout the introduction, please clarify when values mentioned are for horses or humans. It is a bit confusing currently. It is fine to discuss both species in the intro, but clarify.
Line 82: “tournament season” – this is not a common phrase in the horse world, at least not in my country. What does this mean?
Line 91-120: these paragraphs are very short and choppy. I would rewrite to combine some paragraphs and give a better narrative to the paper.
Line 109-110: again, is this in horses or humans?
Line 116: what was the hypothesis of this study?
Line 131-132: did riders complete any sort of survey to assess the fitness of the horse? Would fitness level impact the values of CK and AST?
Line 140-142: were measurements of the weather taken at each event? Data from a wet bulb globe temperature unit would be valuable for this, to help assess thermal stress. Would this have impacted your results?
Line 143: can you define tournament, for those not familiar with it? I would do that at least once in the paper so your audience knows what it means.
Line 143-144: since every horse was sampled a different number of times, was this factored into the statistics run? If so, how?
Line 147-148: here you say there were 20 horses, but in line 140 you say there were 19 horses. Please clarify.
Line 222: please state what significance value was used for data.
Line 224-233: please give p-values and numerical values for this paragraph. Right now it is very vague.
Line 224: I would like to know of your 20 (or 19?) horses how many events each one did. This could be reported as a table, or maybe supplemental? I think it is important to know.
Table1: I struggle to find the relevance of this table. What is the physiological and clinical relevance? Why not show pre/post 30/post 24 hr?
Figure 1a,b,c: the title “Season” is not centered on the x-axis for either figure. Units are needed on the y-axis for both. Were values for multiple events combined within a season? If so, why? Why not show the 5 events separately? I am also confused about timepoint 0 on these graphs. Please explain. For 1C, have you considered different units for the y-axis? Were outliers not removed? That would help clean up the graph. Overall, these graphs all need to be cleaned up a bit.
Tables 2a and 2b: Again, I struggle with reporting the mean values across all measure times. Looking back at the materials and methods – at what time were the echocardiograms performed? Before the events? After? That might help your reader understand these tables. I would update the paper to include this information. Also, you need units for your tables.
Figure 2: what is 0 on the x-axis? Thoughts about separating the 5 competitions?
Line 308: there are many more equine studies that report CK and AST after intense exercise. Please include additional references. This would also give more opportunity for discussion comparing your results to other results, which I think is needed.
Line 311-313: little confused here by the statement “human studies have also shown…” I don’t see a sentence prior to this where there is a similar comparison. It is just mentioned earlier that there is NO evidence in the increase over a season – but there are no references given, which led me to assume that there is no data. But the way the sentence about humans is phrased, it makes it sound like there IS data, it just doesn’t show a connection. This needs clarification.
Line 306-368: the entire discussion only references 4 papers. I think the discussion needs to reference additional studies, certainly in the horse world and perhaps in humans too, for a more robust discussion not centered on speculations. There is a good framework here, but right now the discussion seems sparse to me.
Line 364-368: limitations should be expanded a little bit here. Not only was there a small sample size, not all horses were sampled equally, and only 1 discipline was considered.
Line 370-391: too many choppy paragraphs again. Please try to avoid paragraphs of just 1-2 sentences and condense a bit.
Round 2
Reviewer 2 Report
Comments and Suggestions for Authors
Much improved paper, but still a few edits are needed prior to publication.
Line 329-331: check the number of parentheses. I think you have an extra one in there. Same for Figures 1b and 1c.
Figure 1c: I still have issues with this figure. It’s very hard to read.
Line 385-389: Again, check how many parentheses you have. There looks to be an extra one.
Line418: you should not start a sentence with an acronym. Please do not start the sentence with cTnI.
Line 420-422: this is still not correct. This last sentence about humans still does not fit correctly in the paragraph, and the word “therefore” should not be used. You need to state if in human exercise a cumulative season increase has been shown in CK, AST, and cTnI.
Line 435-436: Do you mean lactate when you say blood values? I do not not know what is meant by “the blood values in this study have not been examined as well.” Please rephrase
General comment: It is not clear from the paper the link between CK, AST and the heart values. It is briefly stated in line 81-82 that they are used to measure performance intensity. You also state in those lines (81-82) that CK and AST are expressed in both skeletal and cardiac muscle, but then in lines 441-443 you state that CK and AST are skeletal muscle enzymes. That seems to contradict what was stated in the intro. So that needs to be fixed. I think the relationship between the heart and CK and AST needs to be clarified in both the intro and discussions.
